# Towards Designing Diegetic Gaze in Games: The Use of Gaze Roles and Metaphors

**Argenis Ramirez Gomez [1,\*]** and **Michael Lankes [2,\*]**

1    School of Computing and Communications, Lancaster University, Lancaster LA1 4YW, UK
2    Department of Digital Media, University of Applied Sciences Upper Austria, 4600 Wels, Austria
*    Correspondence: a.ramirezgomez@lancaster.ac.uk (A.R.G.); michael.lankes@fh-ooe.at (M.L.)

**Abstract:** Gaze-based interactions have found their way into the games domain and are frequently employed as a means to support players in their activities. Instead of implementing gaze as an additional game feature via a game-centred approach, we propose a diegetic perspective by introducing gaze interaction roles and gaze metaphors. Gaze interaction roles represent ambiguous mechanics in gaze, whereas gaze metaphors serve as narrative figures that symbolise, illustrate, and are applied to the interaction dynamics. Within this work, the current literature in the field is analysed to seek examples that design around gaze mechanics and follow a diegetic approach that takes roles and metaphors into account. A list of surveyed gaze metaphors related to each gaze role is presented and described in detail. Furthermore, a case study shows the potentials of the proposed approach. Our work aims at contributing to existing frameworks, such as EyePlay, by reflecting on the ambiguous meaning of gaze in games. Through this integrative approach, players are anticipated to develop a deeper connection to the game narrative via gaze, resulting in a stronger experience concerning presence (i.e., being in the game world).

**Keywords:** Gaze interaction; games; game design; play; eye-tracking; Gaze roles; Gaze metaphors

## 1. Introduction

Gaze-based interactions have found their way into the games domain. Design approaches range from gaze as a tool for highlighting and selecting game entities [1], to camera control [2]. Although the number of gaze-enabled games has continuously risen over the years [3], and its potentials are indicated in other game domains such as Virtual Reality (VR) [4], it is still a niche research activity [5]. One reason for this can be seen in the fact that in many cases, the integration of gaze appears to be superficial. Gaze, in this regard, is considered as being an add-on that is built on top of a fully functional game, leaving it as an optional element.

The first research attempts attempted to gain a better understanding of gaze-based play. Velloso et al. [6], for instance, looked at the use of gaze in games from a feature/mechanics-driven perspective and coined the term "EyePlay" to refer to "those playful experiences that take input from the eyes" [7]. They categorised gaze dynamics by the type of input (discrete vs. continuous), and surveyed state of the art gaze-enabled games to organise them in five kinds of gaze mechanics later. These mechanics are based on the final action triggered by interacting with the eyes, including Navigation; Aiming and Shooting; Selection and Commands; Implicit Interaction; and Visual Effects.

Here, we partly challenge the EyePlay framework to move away from its non-diegetic and mechanics-centred perspective. Here, we refer to the term "diegesis" to indicate a distinction between things that belong to the game story and elements that do not [8]. Therefore, we align this definition to understand how gaze interaction is embedded in the game narrative. We reflect on the ambiguity and metaphors of gaze that could serve as a starting point for designing gaze-based games. Specifically,

approaches such as the EyePlay framework focus on the "formal elements" of a game (e.g., game rules, goals, procedures) [9]. Although these approaches cover a vast area of the design space, they are also limited by the fact that only the game (and not the player) is in the centre of attention. For example, the category "Navigation" deals with gaze as a means to an end to reach a specified location within the game. The player's perspective (i.e., his/her interpretation of the gaze embedded in the game world) plays only a minor role.

It is important to note that we do not cancel the EyePlay framework. We propose a new vision and extend it by introducing a conceptual approach based on the integration of gaze as a narrative layer. Following this line of argumentation, we aim at providing a broader view of gaze-based design in games. We discuss "dramatic elements" [9] (e.g., story, characters and context) that are strongly related to the players and their interpretations of the game world.

A well-known game design model, the "Mechanics, Dynamics, and Aesthetics" (MDA) Framework [10], resembles the integrative character of our approach. After Leblanc et al. [10], the mechanics comprise the formal elements of a game (e.g., resources and winning conditions) that lead to specific player behaviour (i.e., the dynamics of a game). From this behaviour emerges a particular experience, such as having a feeling of being a powerful hero in a fantasy world. It is deemed that dramatic elements also have a significant impact on the players' behaviour and their interpretations of the game world. In sum, we argue for considering dramatic elements in the design of gaze-based games. Specifically, we are interested in gaze as a diegetic element (i.e., something is part of the game world/story) and as a metaphorical figure (i.e., rhetorical and ambiguous concepts of gaze integrated into the game world).

In general, the article contributes to the corpus of research via the following points.

- Introducing the aspect of diegesis in the design of gaze-based interactions in games.
- Reflecting on existing frameworks of gaze-enabled play and extending them.
- Using gaze roles and metaphors to characterise and categorise gaze-based interactions in games.
- Presenting forms of the practical application of the proposed concepts through a design case study.

To reach the goals, the article is based on the following structure: After a state-of-the-art overview (Section 2, "Gaze in Games"), the work addresses non-diegetic design approaches of gaze integration (Section 3 "Non-Diegetic Gaze Interaction"), followed by a description of gaze roles (Section 4 "Gaze Roles") and metaphors (Section 5 "Gaze Metaphors"). Additionally, we apply the proposed framework and present a design case study that illustrates the practical application of the structure (see Section 6, "EyePlay meets Diegetic Design: Twileyed—A Case Study"). Last but not least, the potentials and pitfalls of the approach and the implications of the approach concerning the field of HCI in general are put forward in the discussion section (Section 7, "Discussion").

## 2. Gaze in Games

Eye-tracking is rapidly becoming a popular input device for interaction in games. Overall, gaze is used to augment the feeling of immersion, by controlling visual graphics, or by enhancing other game inputs performance, such as the mouse [11]. To date (July 2019) and according to the primary gaming eye-tracker manufacturer (Tobii), 145 commercial games [3] (including mainstream franchises [1,2,12–15]) integrate gaze interaction in their gameplay. Research games aside, gaze pointing is used as the mechanism for the explicit selection of objects of interest [16], trigger actions [17], moving to where we look [18], or aiming tools and weapons [17,19] around the game scene.

The main thrust of gaze interaction mechanics in games range from the use of the eyes for accessibility [20], to substitute the mouse [21], or to move the character by selecting with gaze the arrows displayed on the screen [22]. Others used where the eyes point to directly move the characters to that position [23], on a horizontal axis [24] or aim their direction [25]. Players can trigger actions by aligning where the eyes point with objects in the game, to select them [26], activate different game defence modes [27] or shoot [28]. Generally, gaze is used with other inputs because we use the eyes

not only for interaction but also as a sensor to perceive the scene. This means that gaze interaction is usually presented with other input modalities [29], to use the eyes to signal interest and voice [30,31], gestures [32] or touch in a multiplayer game [33] to trigger the action. Other gaze-enabled games, use gaze gestures to control the movement of a ball in a maze [34]; smooth pursuit eye movements, happening when we couple the gaze and game object's motion to select targets [35,36]; or blinks to recharge guns [37].

Moreover, implicit gaze interaction has been used to predict the player choices [38], control the camera viewpoint [39], graphics rendering [40,41], imitate realistic eye movements in immersive 3D environments [42] or to adapt the story [43] and difficulty of the game based on the player's gaze behaviour [44,45]. Others used gaze input in games to visualise the gaze point to enable communication between players [46–48] or used it in collaborative [49] and competitive [50–52] games.

Our work follows these examples and aims to create new game dynamics based on the gaze roles and visual metaphors they introduce in the gameplay. In contrast, our approach moves away from a non-diegetic and sensor-based description of gaze mechanics. We propose to leverage a connection between the game story and the use of gaze through narration. We aim to move forward, and design gaze interactions integrated into the game story. Using a diegetic design approach allows introducing gaze interactions that are relevant and without which the game would not make any sense.

## 3. Non-Diegetic Gaze Interaction

One could consider the EyePlay as a survey or a design framework, guiding the possibilities and opportunities that gaze interaction bring in games. On the other hand, this approach can be defined as a non-diegetic approach for design, with a sensor-based view. This means that the gaze mechanic or interaction is not part of the game, but complements it. Gaze pointing in games is used to automatically tag characters that you look at to know whether they are enemies [2,14]; to have cleaner UIs when you are not looking at the game menus [13]; to point weapons towards the opponents that you look [12,15]; or to move the avatar towards the gaze point direction [1]. These augment the gameplay and player experience, but also they could remove the game difficulty.

Overall, gaze-enabled games offer a secondary role for gaze interaction. In other words, if a designer were to remove the eye-tracking device that provides input for interaction, the user would still be able to play the game. The game is not dependent on gaze as this is only substituting, for instance, the device that controls the gameplay or facilitating the interaction during the game. This can be possible because, in the five defined dimensions, gaze is generally used to substitute or augment the game controller, affecting the gameplay experience. Others use it as as a tool, that sometimes models a strategy and affects the player experience (see Figure 1). Therefore we could divide the different gaze mechanics into two non-diegetic uses:

- Gaze for Gameplay: Navigation, Aiming and Shooting, Selection and Commands.
- Gaze as Tool: Implicit Interaction, Visual Effects.

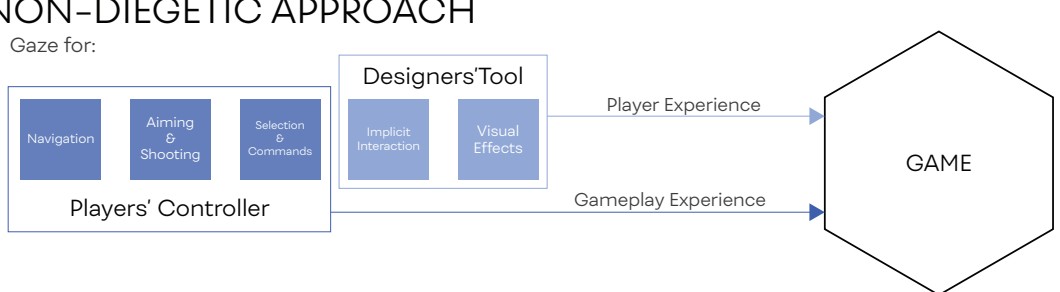

**Figure 1.** Non-diegetic design approach based on Velloso et al.: Survey EyePlay.

*Towards a Diegetic Gaze Design*

Gaze-enabled game examples are populated with applications following this technical approach. State-of-the-art and commercial games use gaze interaction as an extra feature to augment the gameplay experience (complementing controllers) and provide greater immersion.

Here, we analyse the current literature seeking examples that design around gaze mechanics and propose to follow a diegetic approach to design gaze-enabled games. We are interested in the cases that are based on the role of gaze inside the game and the use of gaze-based metaphors to integrate it into the game narrative (Figure 2).

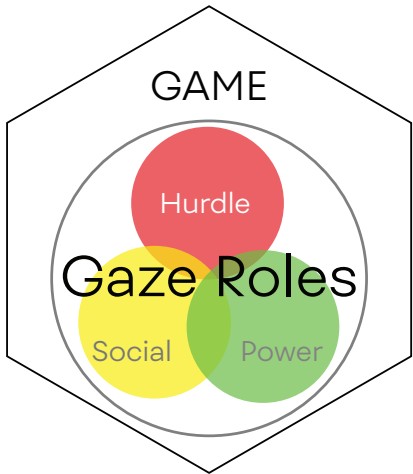

**Figure 2.** Diegetic Design approach with gaze roles.

## 4. Gaze Roles

There are fewer examples of gaze mechanics in games that blend into the gameplay narrative. These follow a diegetic approach, and gaze interaction is part of the game, the means to an end and the artefact that allows the player to trigger specific actions that are in-line with the game narrative. Within the game, gaze has a role and is integrated into the game narrative, making the player necessitate the eye-tracker to guarantee the successful and desired gameplay experience.

Therefore, gaze affects not only the game experience, the narrative or the player experience, but creates a link between both that benefits the overall gameplay. One of the main characteristics of this approach is that if the eye-tracker is removed from the game, the player cannot play and the experience makes no sense. Sometimes the game can be impossible to use or control. When it does, for example, by replacing gaze pointing with mouse pointing or other input devices, the interaction is no longer suitable nor aligned with the game narrative.

For example, in the game "The Royal Corgi" [53], the player, in a first-person view, needs to network with the King's counsellors and win their trust to become the next royal corgi instructor. In the game, the player uses the keyboard to navigate around the room and the dialogue boxes. Gaze influences the reaction of the characters the player communicates with. For instance, they can upset a character if they do not look at them or look distracted by looking somewhere else. Therefore, the game is built to explore the social use of gaze, so wherever the player gaze is pointing, the other characters will modify their answers. Accordingly, if we remove the eye-tracker, the game has no thrill and probably no difficulty as it would be a plain conversation with avatars. If we were to substitute gaze by another pointing input, such as the mouse, mouse-pointing would not fit the interaction effects. The characters would refer to the player visual attention, that would be modelled by the position of

the mouse, creating an ambiguity between the input type and the content or narration of the game for the player.

Thereby, whereas the diegetic gaze mechanics are integrated into the game narrative with more considerable influence, one could consider that gaze has a role within the interaction dynamics and the gameplay. We identified in the literature three typical roles:

- Social Gaze: The use of gaze related to trigger social constructs, behaviours and those events related to human relationships of attention. This role illustrates the gaze mechanics used to interact with characters, e.g., greeting or show attention socially.
- Gaze Power: When gaze is used to trigger actions that would be considered fictional or not possible in real-life. This role represents the gaze mechanics that allow the player to use special features that belong to the game avatar just by looking.
- Gaze Hurdles: The function of gaze assumes a curse or difficulty and a problem to the player posing a game challenge. This role illustrates the gaze mechanics that penalise the player for looking.

These are based on the assumed function or part that gaze interaction plays in the game. Roles can be considered not only the purpose but also the effect or consequences it triggers.

### 4.1. Social Gaze

This role uses gaze as a social artefact. Similar to a live (real) environment, our eyes can indicate social cues inside games. For instance, we can look at people we want to interact with or avoid looking at people that we are not interested in talking to. Generally, the social role of gaze for gameplay interaction is leveraged to provide character awareness and allows the player, for example, to greet other characters on eye contact [1,15,54,55], fostering greater immersion and emotional bonding between characters [56].

However, more complex social behaviours could be performed with gaze. For instance, in Royal Corgi [53], the player's gaze can show disinterest, distraction, respect, interest or even draw the attention of those characters. On the other extreme of social interaction, gaze interaction with a social role could serve to indicate submission by avoiding to look at threatening characters or even use eye winks as a seduction tool [57].

### 4.2. Gaze Power and Player Empowerment

This role uses gaze as a fantasy or make-believe artefact. Gaze interaction is leveraged to provide the player with powers and special abilities that would not be possible to perform outside the game environment. Examples showcase being able to petrify opponents [58], freeze enemies [59,60] or hack other characters in the game [61], just by looking. Others use pupil dilation to control power within the game and to close the eyes to disappear [62] or explicit gaze pointing to discover hidden pieces of the game story [63].

"Power" as a gaze role allows providing players' empowerment in the gameplay. Games illustrating this interaction mechanic offer the player game control beyond the traditional gamepad or other controller, and gives them the ability to highlight objects of interest by having special psychic powers [64], or showcasing the avatar survival skills [15,65]. Unmistakably, we could perform those actions with a different pointing input device, such as the mouse. However, the eyes are a natural attention sensor, and the use of gaze as input for interaction can integrate this power more intrinsically and smoothly.

Moreover, gaze interaction has served not only to give power to the player but also secondary users. Maurer et al. [66] created a version of Super Mario Bros in which half of the game scene was made black (and hidden). In the game, a second player with the previous role of spectator can control with their gaze the position of a hole to see through on the hidden part of the screen, allowing the

primary player to see what is coming next. The described approach used gaze to empower a new player that previously, and without eye-based input in the game, was a mere spectator of the gameplay.

### 4.3. Gaze Hurdles

This role uses gaze interaction as an obstacle, curse, flaw or impediment, making it a challenging artefact. Games using gaze interaction as a burden use rules that penalise the player during gameplay just by looking. The main characteristic of this role is to use gaze pointing against the player and trigger unwanted outcomes. This has been generally exploited in implicit gaze interaction games where the player could trigger sound effects when looking at objects in the scene [67], making the experience more immersing and sometimes scarier [68]. We identified the last two examples as "hurdles" because the gaze effects are unexpected to the player, and they are created to provide them with an "unsettling" and immersive feeling during gameplay.

Others, like "Screencheat" [50] or "Ticket to Ride" [51], play with spoiling the players' gaze behaviour by showing a visualisation of the gaze point to the opponent. Although these dynamics represent a passive and implicit use of gaze interaction, they indeed illustrate the extents of this gaze role.

Explicitly, a gaze hurdle is aligned with dynamics that make the players aware that when they look something terrible can happen, and they need to deal with the consequences. Examples like "VirusHunt" [19] and "SuperVision" [69] present game mechanics that use gaze interaction against the players, making them necessitating to look away from the screen and rely on peripheral vision to be successful in gameplay.

### 4.4. Defining Gaze Roles through Metaphors

Gaze interaction roles can be perceived as ambiguous mechanics in gaze. It might not be evident to the player, for example, that they can look at someone to intimidate them (*Social*), petrify enemies by staring at them (*Power*) or not looking at objects if they want to succeed in the game (*Hurdle*). These three examples could be translated as gaze aiming or selection mechanics from the perspective of the game control, but their effects within the game might be considered confusing.

Take as an example the game "SuperVision" [69], in which the player needs to rely on peripheral vision because they will be penalised if they look. This mechanic can be confusing to the player given that the game is gaze-enabled and relies on eye-tracking technology. Gaze interaction is coupled with looking, and not looking or looking away is perceived as an uncommon interaction. In the game, the authors solved this ambiguity by assigning the dynamic of aversion to pop culture and Greek mythology characters. For instance, the designers included the mythological Medusa, who was able to turn anyone into stone by looking at them.

Therefore, these references or metaphors serve as a figure of narration that symbolises, represents and is applied to the interaction dynamics, helping the player understand the gaze role.

## 5. Gaze Metaphors

Research on metaphors in the context of gaze is carried out in various fields: it ranges from linguistics (e.g., Lorenz [70]), to intercultural research (Sakuragi and Fuller [71]), film studies (e.g., Brakhage [72]) and game studies (e.g., Westecott [73]).

In linguistics, for instance, the notion of metaphor describes how the world is understood and interpreted by humans. This can be seen in expressions referring to gaze and seeing, such as "seeing is touching" or "the eyes are a container for emotion" (Li [74]). In agreement with Begy [75], we see the term "metaphor" not from the perspective of linguistics, but rather in the cognitive sense. Specifically, we are interested in the metaphorical projection, where knowledge obtained from one domain is transferred to another one. In the context of games, this means that metaphors outside the games domain are translated into the game's diegesis. Gaze metaphors in games offer the possibility

to do something that is not possible with traditional scenarios for design, opening the space for novel opportunities.

For example, the concept of the "Midas touch" is based on the well-known story of King Midas and his curse: Everything turns into gold that he touches (even his family is transformed into golden statues). Traditionally, the metaphor conveys the concept of being greedy. In the interaction design domain, it describes the issue that in gaze-based interactions, the perception and the user's actions are not decoupled from the interaction process [76]. In games, the metaphor is employed as either a power or as a curse of the player's avatar. In one case, players might look at objects to turn them into gold and get rich. In another game, the player could take the role of the Midas character whose goal is to avoid looking at specific game entities. The notion of the "Midas' touch" metaphor could also be translated into the social domain: The player has to avoid looking at the non-player characters (NPCs); otherwise, he/she will experience social fear and cannot move.

Sometimes metaphors can help to explain ambiguous or challenging interactions, such as playing with peripheral vision [69] or having to close one's eyes for interaction or pause the game [77]). Furthermore, mechanics can change throughout a play session (e.g., the gaze-based interaction could start as a curse and turn into a superpower). This ambiguity cannot only be introduced by the game through mechanics, but the meaning of gaze could also be defined by the players and/or diegetic elements. For instance, in Lankes et al. [48], gaze was utilised as a tool to augment playful collaborative activities. By introducing various forms of social gaze-based behaviours, a player could renegotiate the meaning of their counterpart's gaze guided by psychological concepts (e.g., joint attention).

*5.1. Social, Power and Hurdle Gaze Metaphors*

We can find examples of different eye-based metaphors used in gaze-enabled games throughout the related work (see Table 1). These metaphors define the gaze mechanic, so it makes sense to the user to demonstrate that it is necessary and integrated into the game narrative. Not using a metaphor to strengthen the gaze role, could make the interaction or game dynamic to be perceived as ambiguous.

For instance, we could justify a social use of gaze, such as to make characters aware that we are looking at them, by using metaphors of "looking to greet" [1], or "Eye Contact" [56]. Other games ("The Royal Corgi" [53]) can explain the fact that gaze is a social pointer and will trigger an possible unexpected reaction from the NPCs by using gaze metaphors related to social behaviour, such as "looking distracted", "looks of interest", "disrespectful looks" or "avoiding to look". In the game, the character with whom the player interacts with explains that the user is disrespectful or that he/she is looking distracted. Without the metaphors, "The Royal Corgi" might provide the wrong experience with unintended outcomes, like characters moving away without any explanation when we do not pay attention to them.

Gaze powers are easily integrated into the game narrative with metaphors. The designers had the opportunity, for example, to illustrate that a gaze selection could "freeze" the enemies [59,60], or even "looks could petrify" them [58]. Without a metaphor, gaze powers would be aligned with gaze pointing for aiming and selection, and thus representative of the game control that affects the gameplay experience. When using a metaphor, gaze becomes an artefact of the game narrative to explain that objects highlight when looked at because you, as the main character in the game, have a "third eye" power [64]; events appear because you can "unravel stories" with your vision when looking around [63]. Moreover, a gaze power metaphor could explain unusual gaze interactions, such as closing one's eyes. In "Invisible Eni" [62], the player can make Eni (main avatar) disappear into smoke to protect herself from the monsters. These could be "Closing you eyes for safety", but also could be described with metaphors of "blind faith", "concentration" or "fear".

Finally, gaze hurdles metaphors justify why gaze is posing a challenge to the player. When the player gets penalised during gameplay for looking at game objects, like in VirusHunt [19] or SuperVision [69], the player might find it ambiguous that they cannot look at the scene. However,

when using metaphors like "looks that infect" [19] or "looks that kill/petrify/charm" [69], it is then understandable for the user, and it makes sense, that when they look, bad things could happen.

Metaphors are usually aligned with a particular use of a gaze interaction "Role" but are not bonded to a unique one. For example, in the "La Rochelle Lab" game [57], a "look that charm" is used to make the player wink at an enemy to get their homework back, hence using gaze with a social role. In the "Narcissus" game in SuperVision [69], the same metaphor is used as a gaze hurdle, as the player needs to sort frames without looking directly or they will fall in love with them.

Similarly, in SuperVision [69], "looks that petrify" are used as a gaze hurdle when Medusa, one of the game characters, look at mushrooms she needs to remove, turning them into stones. In "Medusa's Lair" [58], the same metaphor is used as a power in a multiplayer scenario. In the game, one player acts as Medusa and can petrify the opponents by looking at them while they run around trying to escape her. Moreover, but not specifically in a computer game,"Medusa's gaze" could act as a social gaze metaphor, like in the children game "Red Light, Green Light". In the game, all players must stop and pretend they are statues (petrified) when the leading player (opponent) turns around to look at them, or they will be disqualified. Whereas this could be considered as a power or a hurdle for the players, they choose to be immobile, and it is the intimidating look of the opponent that makes them stop.

In effect, gaze roles and metaphors form a complex relationship between them, influencing each other retrospectively and allowing the creation of different game dynamics.

**Table 1.** List of Surveyed Gaze Metaphors related to each gaze role.

| Metaphor | Gaze Role | Game Examples |
|---|---|---|
| *Looking Distracted* | Social | The Royal Corgi [53] |
| *Looks of Respect* | Social | The Royal Corgi [53] |
| *Submissive Looks* | Social | The Royal Corgi [53], La Rochelle Lab [57] |
| *Avoiding to Look* | Social | The Royal Corgi [53], La Rochelle Lab [57], SOMA [78] |
| *Looks of Interest* | Social | The Royal Corgi [53] |
| *Look to Greet* | Social | Assassins' Creed [1], Tomb Raider [15], |
| *Looks that Calm* | Social | Wayla [79], Spectrophobia [80] |
| *Eye Contact* | Social | Agents of Mayhem [54], Shelter 2 [56], Reflections [81], Knee Deep [55] |
| *Looks that Freeze* | Power | Limus and the Eyes of the Beholders [60], Biofeedback [59] |
| *Not Looking for Safety* | Power | Invisible Eni [62], Nevermind [77] |
| *Having the Third Eye* | Power | The Channeler [64] |
| *Having an Eye for Detail* | Power | Amphora [82] |
| *Look to reveal* | Power | Fractile [63], Mario Bros [66], Tomb Raider [15], Paws [65] |
| *Hack at Gaze* | Power | Watch Dogs 2 [61] |
| *Survival Instinct* | Power | Tomb Raider [15], Paws [65] |
| *Looks that Charm* | Social, Power | La Rochelle Lab [57], SuperVision [69], The Royal Corgi [53] |
| *Starring Competition* | Social, Power | The Revenge of the Killer Penguins [83] |
| *Looks that Intimidate* | Social, Power | The Royal Corgi [53], Shynosaurs [84], Spectrophobia [80] |
| *Blind Faith* | Social, Power | Invisible Eni [62] |
| *Looks that Infect* | Hurdle | VirusHunt [19] |
| *Looks of Fear* | Hurdle | La Rochelle Lab [57], Glimpse of Fear [68], Medusa's Labyrinth [67] |
| *Disrespectful Looks* | Social, Hurdle | The Royal Corgi [53] |
| *Looks that Defy* | Social, Hurdle | Spectrophobia [80], SOMA [78] |
| *Looks draw Attention* | Social, Hurdle | Keyewai [85], Dying Light [86], Screencheat [50], Ticket to Ride [51], Mills [50] |
| *Looks that Kill* | Power, Hurdle | SuperVision [69] |
| *Looks that Petrify* | Power, Hurdle | SuperVision [69], Medusa's Lair [58] |
| Medusa's Gaze | Power, Hurdle | SuperVision [69], Biofeedback [59], Medusa's Lair [58] |

### 5.1.1. Social Powers

Some social gaze behaviours can be perceived as powers. In "Amphora" [82], the player needs to solve puzzles and unfold the character story through connected tales. In the game, the player with "an eye for detail" can make butterflies fly away. Later, players can see them on the menu screen where they show the player's progress in the game. In this case, gaze could have a social role because the butterflies are aware of the player's attention on them. However, it could be a power because they sense the player's gaze and magically fly away to another scene. Accordingly, a "Staring Competition", in "The Revenge of the Killer Penguins" [83] can represent a social gaze metaphor to require the player to continually look to win, but also could be considered the power of submission.

In "Shynosaurs" [84], players need to save the "cuties" by mouse-dragging them inside a fenced area. While solving this task, the enemies ("shynosaurs") will try to take the wandering "cuties" away. However, the player can look at the enemies to intimidate them to make them shy and stop their attack. The longer the player looks at the shynosaurs, the shyer they become until they run away crying. This game illustrates using a social metaphor to represent both a social and power gaze interaction. Gaze is used to make the enemies stop, freezing them on the spot by staring at them (power). Nevertheless, as the player keeps their gaze on the target, the social gaze intimidation is adopted.

### 5.1.2. Social Hurdles and Powerful Hurdles

Social metaphors of gaze attention are usually presented in games for characters awareness. In gameplay, when the player stares at an NPC, this will generally look or even wave at the user [1]. However, in "SOMA" [78] and "Dying Light" [86] this social role is turned against the player. In both games, enemies (robots and zombies respectively) will become aware of the player's gaze and attack them, creating a social gaze hurdle.

A "Look that kills" might be considered a gaze power metaphor, because it is not realistic, makes no sense and only works with its metaphorical meaning. However, depending on the game narrative, this gaze mechanic could also be considered a *Gaze Hurdle* because it has deadly consequences. In the "Cyclops" game in SuperVision [69], this metaphor is used as a gaze dynamic. The player needs to sort balloons by manipulating them with the mouse in their peripheral vision. If they look at the balloons, they will explode and fall, making gaze a hurdle and posing the challenge. On the other hand, in the game, there are also bad balloons the player needs to destroy, therefore gaze turns into power, and constantly switches between the two roles.

### 5.1.3. Socially Powerful Hurdles

In a two-player game like "Keyewai" [85], gaze is used to steer the avatars' torch direction. During the game, players need to explore a forest inhabited by cannibals. If the torch is pointed at the cannibals, they will become aware of the player and attack them. However, if both players light-point at them, the cannibals will get confused and do not know whom to attack. The social gaze metaphor "looks draw attention" is used in this game as a hurdle. When the cannibal detects the players' attention, it creates an unwanted event, the attack. Nevertheless, the collaboration between players can transform this gaze mechanic into a gaze power, when "drawing somebody's attention" turns into a distraction/confusion strategy.

"Spectrophobia" [80] showcases the use of a gaze metaphor that is transformed depending on the gaze role determined by the game narrative. In the game, the player acts as a young man who attempts to find his wife in an abandoned mansion. He is haunted by invisible and sanguinary monsters who can only be seen through his reflection in mirrors. The monsters have different behaviours that the player affects through "Eye Contact". Some monsters attack when looked at (Hurdle: *Looks that Defy*"), other are pacified (Power: "Looks that Calm"), whereas spiders will run away when aware that they are aware of being gazed (Social: "Looks that Intimidate").

These examples showcase how the combination of different roles and metaphors create a constellation of examples broadening the opportunities to develop novel eye-based interaction mechanics in gaze-enabled games.

## 6. EyePlay Meets Diegetic Design: Twileyed—A Case Study

"Twileyed" is a collection of three mini-games that explores the use of ambiguity in the game rules and tension in gaze interaction for Selection, Aiming and Navigation. The games combine a representation of three of the main non-diegetic uses of gaze with gaze roles and metaphors, making gaze interaction necessary in the gameplay.

The three games include popular characters from fiction that serve as the first step in the use of metaphors. The games consist of a collection task with challenging interactions. The keyboard arrows are used to move the characters and gaze input is used to modulate navigation by selecting characters that the players want to move, aiming the character's direction or shooting. The games are designed to introduce incompatible and ambiguous contexts for gaze interaction by using gaze metaphors.

### 6.1. Dorian's Pictures—Eye Selection and Social Gaze

In this first game, gaze is used to select the character the player wants to move. The game story is based on the literary character of Dorian from "The Picture of Dorian Gray" by Oscar Wilde. Although Dorian is immortal, he could be killed if he is in front of his picture. The game objective is to move Dorian to collect pieces of his picture in a party populated by "Guests", "Servants" and "Assassins" that try to kill him. "Servants" and "Guests" can stop the Assassins by getting in their way.

To move Dorian the player needs to look at him because he is very narcissistic and requires that all the attention is on him. On the other hand, both guests and servants move at the same time and direction that Dorian does (or attempts if he does not have any attention), because they admire him. Moreover, if Dorian (the player) looks at either of them, they will feel intimidated and stop moving (see Figure 3, left). This is helpful to avoid they collect a piece of the frame.

This game showcases the use of several gaze metaphors with a social role, combined with interaction mechanics for gaze selection. Selecting Dorian takes a social role to promote "looks of attention" and comply with the narcissistic behaviour of Dorian. On the other hand, when the player looks at the other characters, it is translated to the metaphors of Dorian looking at them with a "look that intimidates" and makes them "paralyse" or "freeze".

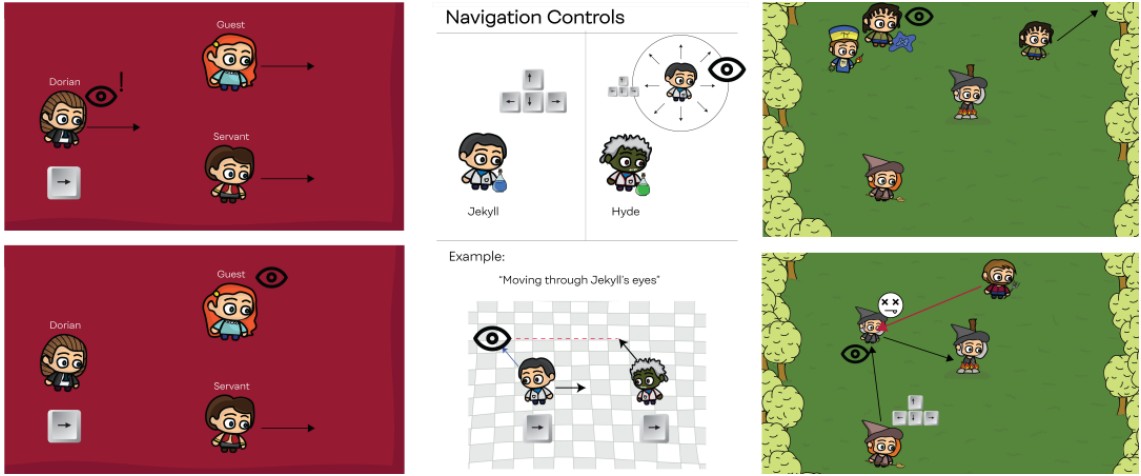

**Figure 3.** *Twileyed*'s three games' rules. (**Left**) "Dorian's Pictures" based on Selection and Social Gaze. (**Center**) "Jekyll and Hyde" based on Navigation and Gaze Power. (**Right**) "The Witches" based on Aiming and Gaze Hurdles.

### 6.2. Jekyll and Hyde—Eye Navigation and Gaze Powers

In the second game, gaze is used for Navigation to point at the direction of the avatar's movement. The game story is based on the story of "Dr Jekyll and Mr Hyde" from "Strange Case of Dr Jekyll and Mr Hyde" by Robert Louis Stevenson. In the fiction, Dr Jekyll would transform into his alter ego Mr Hyde by drinking a special serum, but in the game, Dr Jekyll managed to separate himself from Mr Hyde and both need to drink a serum to keep it that way. Therefore the game objective is for each to collect their appropriate serum bottles, Jekyll the blue ones, and Hyde the green ones without picking up each others'.

Due to a side effect, both characters will move at the same time, using gaze to point towards the direction of the movement. However, whereas Dr Jekyll moves according to the arrow keys, Hyde moves "through Jekyll's eyes". This means that Hyde will move when Jekyll moves towards the direction the second is looking at (see Figure 3, centre). This makes Jekyll the centre of an invisible joystick to point at the direction of the other character's movement. For example, if the player looks to the top-right and moves Jekyll down, he/she will go down, whereas Hyde will go to his top-right.

The game challenges the player to understand a biased gaze mapping for navigation but introduces a gaze metaphor that could be considered a power. The player can move both characters at the same time, because they are "connected with a side effect", and gives the user the empowerment to move Hyde "through Jekyll's eyes". This is presented as a side effect but could also be seen as another metaphor to describe "looks that draw attention" as we could see where Jekyll looks like the reference position that Hyde will mimic and follow.

### 6.3. The Witches—Gaze Aiming and Hurdles

Finally, in the third game, gaze is used for "Aiming" to shoot where the player looks when a key is pressed. The game story is based on the fictional figures of the "Witches" and a witch hunt during the "Witches Trials". In the game, the player, controlling a "witch", needs to collect other witches to save a captured fellow witch. During the task, there are Villagers hunting for witches and trying to burn one at stake. To fight them, the player can cast water spells where she looks to repel the enemies. However, magic comes with a price and villagers can detect where the witch is looking and go towards that position.

Therefore, during the game, the player will spoil their strategy while aiming with gaze. Partly, this represents a non-diegetic use of gaze aiming for shooting, but also implicit gaze interaction while revealing where the eyes point within the game. This challenges the player to solve the task with this gaze hurdle based on metaphors aligned with "the price of magic". This narrative can also explain with *"looks that draw attention"* or "looks that inform" and "spoil", or even giving the villagers the power of "the third eye".

In a nutshell, the game Twileyed showcases how considering a diegetic use of gaze roles and metaphors for interaction in game design is compatible with previous frameworks like the EyePlay. Moreover, they can work together to offer diegetic interactions and novel gaze mechanics opportunities.

## 7. Discussion

Our approach provides multiple benefits for designers by introducing a novel perspective of game design in the context of gaze-enabled games. Instead of implementing gaze as an additional game feature, we propose the integration of gaze through metaphors and roles that are embedded in the narrative structure of a game. In doing so, we could extend existing frameworks, such as the EyePlay framework, by focusing how gaze is embedded in games. Following this notion, we anticipate that through this integrative approach, players might develop a deeper connection to the game narrative, resulting in a stronger experience concerning presence (i.e., being in the game world [87]). As mentioned in previous sections (Section 5 "Gaze Metaphors"), we are particularly interested in metaphorical projections, where metaphors are transferred from one domain to another.

Games in this regard are relevant because they could not only depict everyday-life situations but also grant players to explore environments and (social) situations that would not be possible in reality. Thus, games could be regarded as opportunities for innovative designs, where established metaphors could be modified, or new ones be introduced.

### 7.1. Applying the Framework: "Blind Faith"

Here, we discuss how designers could benefit from using the proposed framework by unfolding a gaze metaphor into different gaze interaction mechanics and roles. To illustrate the process, we decided to use the metaphor "blind faith". Traditionally, the phrase "blind faith" describes a belief without real understanding or questioning. Technically, "blind faith" could be considered when the user closes the eyes and the sensor, the eye-tracker, has no input.

We understand this metaphor to refer to the act of not looking with three different meanings according to each gaze role. Socially, "blinded faith" refers to not looking because we trust, for example, a narrator that guides us through the action of the game. As a power, "blind faith" can be shaped into the action of closing the eyes and trust your inner powers. For example, superheroes might shut their eyes to concentrate on their power, e.g., teleporting, or disappear into smoke [62]. If we consider this metaphor a hurdle, we could imagine a game that uses when we do not look at the game action to penalise the player.

If the designer wants to implement gaze navigation, this metaphor could be used not for the action of moving per se, but to trigger safer motion. We could imagine a game that presents the players with a field filled with underground mines that they need to cross. The players are accompanied by a narrator that knows where the bombs are, requiring them to follow her advice. This presents a dynamic based on social blinded faith, in which the player needs to trust the narrator to move around. Gaze could be used to move towards where the eyes point, according to the narrator. Moreover, gaze interaction could be used to make the player close their eyes. The first example requires to follow precise instructions an relies on good game story design. Developers could implement the latter by deactivating the bombs when the eyes are closed, showing the player's "blind faith" on the narrator.

When using the metaphor for selection, we could think of a game scene in which the players are offered to choose objects they cannot see. In this situation, the player might need to select with gaze objects that are hidden (in a black box). On the other hand, the players could also close their eyes and trust other senses, such as spatial sound. This dynamic can introduce choosing between two objects relying on something that was not available when the players were looking. For gaze aiming and shooting, we could create a fictional scenario where the players have special powers that are only available when they close their eyes. Moreover, in a Virtual Reality game, we could use where the head points to trigger this power. We could think of this situation in which the gaze is shut and head points where we want to fire. On the other hand, we could design a game that to shoot, the player needs to turn around and not look, having a blind faith that the shoot will reach the enemy.

Finally, if designing with implicit gaze in mind, we could imagine a game that uses not looking with the blinded faith meaning to trigger something else. It could be, for example, the action to make the avatar rest. This could help to immerse the player in the experience, but also transform the implicit interaction into explicit once the player realises that closing the eyes triggers that effect. Accordingly, we could anticipate two effects of this metaphors in gaze-enabled games. First, a potential enhanced and augmented immersing game experience. Nevertheless, further research is necessary to establish this diegetic approach could create such an effect on the player experience. Second, a diegetic use of gaze interaction in the gameplay.

### 7.2. Contextualizing the Framework

Our results are aimed at contributing to the vast body of research projects and contributions in the field of uni- and multi-modal interactions. We are aware that gaze as a modality in the context of interaction design is well-studied outside the games domain (e.g., [88,89]). Research in this context

ranges from Human-Robot Interaction [90] to the automotive context [91]. We deem that the inclusion of narrative elements driven by playful concepts could provide insights for novel interaction designs. Games and playful environments are especially suitable. They allow physical activities through various in- and output devices (i.e., Kinect [92], Leap Motion [93]), and they often facilitate the introduction of different interaction roles (Lankes et al., 2017) embedded in metaphors. Furthermore, games are a useful instrument in the context of nonverbal interactions: For instance, games have been employed to investigate the impact of facial expressions on subjects [94].

We also see potential connections between metaphors in the context of gaze-enabled play and the concept of "Embodied Interaction". Following the argumentation of Maurer et al. [95], Embodied Interaction deals with designing systems that are meaningful for users, which are interpreted as social beings with sentient bodies. Through their physical attributes and their senses, people experience their surroundings [96] that guide their actions and cognitive processes [97]. By including gaze as a narrative element in the form of metaphors, players could interpret and influence a given game situation through their bodily senses.

The term "Embodied Interaction" also implies nonverbal communication in general (e.g., body postures, facial expressions, hand gestures and gaze) as well as the aspect of uni- and multi-modality in interaction design. Consequently, we are the opinion that our considerations could be applied to other modalities, such as gestures, facial expressions, etc. For instance, our views could contribute to the work of Djordjevic [98]. The researcher investigated established HCI metaphors and created guidelines for identifying metaphors that aimed at maximizing the clarity and reducing the users' mental workload. By addressing the aspects of narrative elements and story, we could extend the framework as it exclusively focuses on functionality and visualization (i.e., aesthetics and consistency). Furthermore, the considerations made in this article could also be applied to multi-modal constellations, as gaze has been studied in combination with gestures [99], facial expressions [100], or voice [101].

### 7.3. Limitations

Although our set of gaze metaphors appears to be promising, some limitations have to be acknowledged. Similar to other categorisations, such as the MDA framework [10], the proposed set of gaze metaphors and roles are based on a limited number of game examples. Furthermore, the game examples show that the gaze metaphors are not independent of each other (i.e., one gaze metaphor could require/exclude another metaphor). For instance, the metaphor "Submissive Looks" could be strongly related to "Avoiding to Look" in a situation where a NPC shows signs of submissive behaviour by avoiding to look at the player's avatar. Another limitation can be seen in the fact that other (non-)verbal communication channels, such as gestures, head posture, or facial expressions, are not addressed. These signals could have an impact on the way how a gaze metaphor is perceived. This aspect could be relevant in social situations when gaze behaviour and verbal utterances contradict each other. Games, like L.A. Noire [102], use these elements of in-congruent behaviour to communicate if an NPC is lying or is telling the truth. Based on the NPCs performance, the player's avatar, a detective, has to find out if a suspect speaks the truth or is telling lies. On the other hand, multiple channels of nonverbal behaviour could emphasise a specific metaphor through consonant constellations. Characters might employ a greeting gesture via gaze and wave their hands to underline the meaning of a metaphor. An example that directly uses these combinations can be seen in the game Assassins Creed Origins [103]. In the game, the avatar is being greeted through a hand gesture, when the player looks at an NPC for some time.

### 7.4. Future Directions

We deem that the proposed approach offers several future research directions. The investigation of other nonverbal communication channels forms one direction (e.g., hand gestures, body postures, facial expressions). Our perspective allows us to see acts of nonverbal behaviour as narrative figures. They have the potential to be a crucial element of the game's story and the player's interpretation of

the game world. Metaphors of nonverbal communication in games cannot only be investigated in isolation but can also be observed in combination with nonverbal utterances (e.g., gaze in conjunction with hand gestures or tone of voice). Furthermore, we are interested in continuing our studies of using gaze as an implicit output modality for guiding the player's attention (see ARGENIS WORK and [104] for further detail). The investigation of gaze metaphors in games from an intercultural perspective could also yield relevant results. We are aware that western culture influences our observations and most of the available literature. Comparative studies between different cultures could shed some light on commonalities and differences in gaze metaphors in games.

## 8. Conclusions

This article introduced a diegetic perspective by adding gaze interaction roles and gaze metaphors. Through the analysis of the current literature and game examples, a diegetic approach consisting of game roles and game mechanics was introduced. We presented a list of surveyed gaze metaphors related to each gaze role and showed the potentials of the proposed approach through a case study. These categorisations could support designers in integrating gaze not only as an add-on to existing game design but as an integrative element of the game world and its narrative. Following this notion, our ideas are aimed to extend current approaches by taking the ambiguous meaning of gaze in games into account. We believe that our endeavours are just the beginning of reflecting on the design potentials of gaze and other nonverbal interactions in games.

**Author Contributions:** Conceptualization, A.R.G. and M.L.; Data curation, A.R.G.; Formal analysis, A.R.G.; Investigation, A.R.G. and M.L.; Methodology, A.R.G. and M.L.; Project administration, A.R.G. and M.L.; Resources, A.R.G. and M.L.; Visualization, A.R.G.; Writing—Original draft, A.R.G. and M.L.; Writing—Review & editing, A.R.G. and M.L.

**Funding:** This research received no external funding.

**Conflicts of Interest:** The authors declare no conflict of interest.

## Abbreviations

The following abbreviations are used in this manuscript.

| | |
|---|---|
| MDA | Mechanics, Dynamics and Aesthetics |
| NPC | Non-Player Character |
| VR | Virtual Reality |

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
