# Peer review of "Towards Designing Diegetic Gaze in Games: The Use of Gaze Roles and Metaphors"

_mti, doi:10.3390/mti3040065_

Round 1

Reviewer 1 Report

A big job have been done analysing gaze-based games. The paper has a long list of references.

An interesting approach of classification games into non-diegetic versus diegetic was proposed.

About how the research could be extended. Technical characteristics of gaze-trackers could be analysed for every game group. Nowadays there are known open source gaze tracking projects based on web-camera. Gaze-trackers are expensive devices. If to use web-camera  based gaze-tracker is enough to play some games, these games players auditory could be significantly increased. 

Progress in computer vision and deep neural networks opened opportunity to evaluate user emotions from face image and voice input in real time. Therefore it is interesting how gaze-tracking based games could benefit from combining gaze input with emotions input. I think that such approach i first order is promising for "social gaze" games . 

Author Response

We want to thank the reviewers for taking the time to read our work and for providing their feedback. We addressed in the latest version of the manuscript some of the concerns shared in the reviews regarding the context, framing, presentation and other clarifications.

The framing of the proposed work against general HCI and Multimodal Interaction context and general Discussion.

We extended our work Discussion to reflect on the position of the proposed design framework in past literature in HCI and Multimodal Interaction. We decided to explicitly address how the contribution is related to the journal field and not in the Introduction section. We did not want to draw the attention of the reader away from the proposed framework by referring to the vast history of HCI/MTI in the introduction but to further discuss it in the Discussion section only. Moreover, in the same section, we included a discussion on general HCI metaphorical projections to highlight the relevance of the proposed work and clarified the framework contribution.

Presentation

We clarified the meaning of “diegesis” (widely used in the proposed work) in the introduction (lines 31-32) to address an early definition and help the readability of the article. Moreover, we modified the consistency of our wording to make the contribution more rigorous, e.g. “gaze-enabled games” rather than “gaze-based games”. We edited, as per reviewer suggestions the presented figures to clarify them.

Other clarifications

We clarified the classification for some of the references in the design space to better explain why they were included in their specific “gaze role”. Concretely, we clarified in lines 204-296 the rationale behind considering the trigger of sound effects by gaze as “gaze hurdles”. Finally, we corrected missing references and other general clarifications, e.g. understatement about the existence of different metaphors and definition of the framework in the Discussion section.

We want to thank again the reviewers for considering for publication in the Special Issue Novel User Interfaces and Interaction Techniques in the Games Context in the Multimodal Technologies and Interaction journal.

Sincerely,

Argenis Ramirez Gomez, MSc

PhD candidate, School of Computing and Communications, 

Lancaster University

Reviewer 2 Report

The authors aim at providing a conceptual framework for facilitating the design of computer games that make use of gaze as one of their input modalities.

The proposed "diegetic" design framework is presented as contrasting with a more classical game-structural/input mechanism centric perspective on using gaze examplified by the EyePlay framework found in literature (Velloso et al., 2015), contrasting by instead focusing on how gaze can be integrated into the actual storytelling of the game.

General comments:
The aim is timely as gaze trackers have become ubiquitously integrated in interactive devices in the recent 5 years and the attempts to make use of them in game design can probably be described as being not very systematic.

The overall impression is that the work can be useful for researchers in multimodal interaction and specifically those involved in designing digital games. The review of the gaze-enabled games has a value even on its own. There is however a significant lack of framing of the proposed design framework in past literature in the fields of HCI and Multimodal Interaction. If the authors improve the paper in this regard (which can be done quickly if an experienced HCI researcher is involved) the paper can become really interesting for the audience of this journal.

The motivation for using the greek term "diegesis" for naming the framework is weak and the lack of a proper definition early in the article makes the reader more confused than enlightened. Why not use some variation of the word "storytelling" instead?

Since this journal is _not_ an eyetracking or digital games focused journal, but a multimodal interaction journal, this reviewer finds the framing of the work rather narrow. It would be suitable to more explicitly refer to the vast and long history of Human-Computer Interaction and its sub field Multimodal Interaction which to a large degree can be regarded as disciplines driven by the appearence of new input and output devices in regular and nonäregular intervals. For instance, the pixelated display, the computer mouse, wearable accelerometers, depth cameras have all appeared on the market and subsequently been studied and reflected on, frameworks have been proposed, and some best use case scenarios together with existing interaction modalities identified.

The authors are sometimes a bit sloppy in mixing up "gaze-based games" with "gaze-enabled games" and vice versa. Example: row 55: isn't "gaze-enabled play" more adequate than "gaze-based"? Instead, "gaze-based" would be very applicable for denoting what is described on rows 129-130 but is _not_ used(!)

It is very hard to systematize a complex design space such as gaze-enabled games using a limited numbers of "roles" and "metaphors". I find most of the authors work classifyng phenomena in existing games accordingly as quite plausable. I find it hard to agree on seeing what is described on row 199-201 as "hurdles" though.

In the future work section I miss the possible investigation of using gaze as output modality, in the sense that the game unconsciously or consciously directs the gaze of the player towards certain areas of the game scene.

Detailed comments
-----------------
The abstract illustrations fig 1 and in particular fig 2 are a bit confusing to this reviewer. Fig 1 because it is not clear for whom gaze is a "tool". The game designers perhaps? Or the players? Is "tool" even a good word for what you want to say? In fig 2 I don't understand why you simply not put the names of the three metaphors directly in the left circle and save up some space. What is the arrow for?

Table 1 is very useful and easy to get, in contrast to the above mentioned figures.

There is a missign reference on row 327.

The text on rows 411-413 seems misplaced.

row 419: "by focusing on the ambiguous meaning of gaze in games" -- I don't think this is what you are doing. I find many of the examples you list in your roles/metaphor framework as non-ambigous use of gaze. Clarification needed I think.

row 423: metaphorical projections are very powerful design tools indeed. I miss some references to how metaphors have been used heavily in past HCI work, including how they influenced the famous "desktop metaphor" for the first computers using digital displays.

row 470: "other metaphors might exist" -- well of course! This is a huge understatement. I'd remove this sentence to not look too naive.

Author Response

(The authors gave the same response as above.)
